# Structural Characterization and Heparanase Inhibitory Activity of Fucosylated Glycosaminoglycan from *Holothuria floridana*

**DOI:** 10.3390/md19030162

**Published:** 2021-03-18

**Authors:** Xiang Shi, Ruowei Guan, Lutan Zhou, Zhichuang Zuo, Xuelin Tao, Pin Wang, Yanrong Zhou, Ronghua Yin, Longyan Zhao, Na Gao, Jinhua Zhao

**Affiliations:** 1School of Pharmaceutical Sciences, South-Central University for Nationalities, Wuhan 430074, China; sx20200807@163.com (X.S.); ruoweiguan13@163.com (R.G.); zuo19951224@163.com (Z.Z.); txl6361@163.com (X.T.); wangpin1994@163.com (P.W.); zyr17577300547@163.com (Y.Z.); yinronghua@mail.kib.ac.cn (R.Y.); zhaojinhua@mail.kib.ac.cn (J.Z.); 2State Key Laboratory of Phytochemistry and Plant Resources in West China, Kunming Institute of Botany, Chinese Academy of Sciences, Kunming 650201, China; zhoulutan@mail.kib.ac.cn; 3Institute of Marine Drugs, Guangxi University of Chinese Medicine, Nanning 530200, China; 4National Demonstration Center for Experimental Ethnopharmacology Education, South-Central University for Nationalities, Wuhan 430074, China

**Keywords:** *Holothuria floridana*, fucosylated glycosaminoglycans, oligosaccharide, chemical structure, heparanase

## Abstract

Unique fucosylated glycosaminoglycans (FG) have attracted increasing attention for various bioactivities. However, the precise structures of FGs usually vary in a species-specific manner. In this study, HfFG was isolated from *Holothuria floridana* and purified by anion exchange chromatography with the yield of ~0.9%. HfFG was composed of GlcA, GalNAc and Fuc, its molecular weight was 47.3 kDa, and the -OSO_3_^−^/-COO^−^ molar ratio was 3.756. HfFG was depolymerized by a partial deacetylation–deaminative cleavage method to obtain the low-molecular-weight HfFG (dHfFG). Three oligosaccharide fragments (Fr-1, Fr-2, Fr-3) with different molecular weights were isolated from the dHfFG, and their structures were revealed by 1D and 2D NMR spectroscopy. HfFG should be composed of repeating trisaccharide units -{(L-FucS-α1,3-)d-GlcA-β1,3-d-GalNAc_4S6S_-β1,4-}-, in which sulfated fucose (FucS) includes Fuc_2S4S_, Fuc_3S4S_ and Fuc_4S_ residues linked to O-3 of GlcA in a ratio of 45:35:20. Furthermore, the heparanase inhibitory activities of native HfFG and oligosaccharide fragments (Fr-1, Fr-2, Fr-3) were evaluated. The native HfFG and its oligosaccharides exhibited heparanase inhibitory activities, and the activities increased with the increase of molecular weight. Additionally, structural characteristics such as sulfation patterns, the terminal structure of oligosaccharides and the presence of fucosyl branches may be important factors affecting heparanase inhibiting activity.

## 1. Introduction

Sea cucumber has long been used as food and folk medicine, particularly in some parts of Asia [1]. It is rich in bioactive components such as proteins, collagen, saponins and sulfated polysaccharides [2]. Various studies have shown that these components possess multiple biological and pharmacological activities, such as anti-thrombotic, anti-cancer, antioxidant, anti-inflammatory, anti-bacterial, anti-diabetic, anti-obesity and anti-angiogenic activities [3,4,5].

Fucosylated glycosaminoglycan (FG), a distinct glycosaminoglycan derivative found up to now exclusively in sea cucumbers, generally possesses a chondroitin sulfate-like backbone, fucose side chains and a high degree of sulfate substitution [6]. FG has numerous functions, such as anticoagulant, antiviral, anti-tumor, anti-inflammatory and anti-human immunodeficiency virus (anti-HIV) activities [7,8]. It is worth noting that the potent anticoagulant properties of this glycosaminoglycan derivative selectively inhibit the intrinsic factor tenase (iXase), which may open a new way for the development of antithrombotic drugs [9,10,11]. The intensity of biological activities mainly depends on FG structures, especially on the degree and pattern of sulfation of the backbone and branches, while the structure diversity of native FG is much higher and species-specific [12,13]. Several research groups have been searching for FGs from various sea cucumber species, investigating their bioactivities and developing them as functional food or medicines. As a bioactive polymer, structural variations of FG focus on the amount and position of branches, as well as on the degree and pattern of sulfation of the backbone and branches [8,14,15,16]. Therefore, the structural elucidation of FGs from different sea cucumber species could provide an important basis for the study of their structure-activity relationships.

*Holothuria floridana* is a tropical Atlantic species, mainly distributed in southern Florida, USA. Previous studies on *H. floridana* mainly focused on its ecological distribution, reproductive cycle, etc. [17] Recently, a fucosyl glycosaminoglycan has been reported. According to the structure analysis of low-molecular-weight products from hydrothermal depolymerization by HILIC-ESI FT-MS [18], it was found that its fucose side chains were connected to GalNAc, and not all of the fucose side chains were linked to O-3 of GlcA. The glycosidic bond can be hydrolyzed and broken under acidic conditions, and by controlling the concentration of acid, temperature and time to achieve different levels of acid hydrolysis. Generally, under mild acid hydrolysis conditions, the glycosidic bonds in branches are easier to break than those in the main chain, and the glycosidic bonds of pentoses are easier to hydrolyze than the glycosidic bonds of hexoses [19,20]. Therefore, the hydrothermal depolymerization of fucosylated glycosaminoglycans isolated from *Holothuria floridana* (HfFG) may result in destroying the Fuc side chains. In addition, since this method may lack glycosidic bond cleavage selectivity, various types of glycosidic bonds can be cleaved, resulting in complex and diverse types of oligosaccharides in low-molecular-weight products, which is not conducive to the isolation of pure oligosaccharide compounds. The above reasons may cause differences in results.

The development of chemoselective depolymerization methods could facilitate the progress in the structural analysis of FG from various species. Recently, the deaminative degradation [21,22] and β-eliminative depolymerization methods were used to investigate the exact structure and the structure-activity relationship of FG, and these strategies have been used in FG from *Stichopus japonicus* [22], *Stichopus horrens* [23], *Bohadschia argus*, *Holothuria coluber* [24], *Actinopyga miliaris*, *Holothuria albiventer* and *Stichopus variegatus* [25]. The structure exploration of FG through the bottom-up strategy based on the structural analysis of purified oligosaccharides from chemoselective depolymerization methods has proved useful in confirming the structure of the CS-like core and sulfated fucose branches.

Some sulfated polysaccharides like heparin and CS-E showed heparanase inhibitory activity and could inhibit experimental metastases in animal models [26,27]. Heparanase is a mammalian endo-d-glucuronidase that cleaves heparan sulfate (HS) chains. Heparanase-mediated degradation of heparan sulfate (HS) is the critical process for tumor angiogenesis and metastasis, and heparanase has thus become an attractive target for cancer research. As structurally related to HS, native FGs have showed anti-tumor activity. Low-molecular-weight FG from *Cucumaria frondosa* could significantly inhibit mouse Lewis lung carcinoma growth and metastases in a dose-dependent manner [28]. The latest research showed that the native fucosylated glycosaminoglycan from *Holothuria fuscopunctata* and its depolymerized products have the ability to directly bind with heparanase as a heparanase inhibitor, leading to a significant inhibition of heparanase activity [29]. Therefore, combined with the anticoagulant activity and the low bleeding tendency of dFG and its oligosaccharides, the use of oligosaccharides may have a better effect on tumor patients with thrombotic tendency.

In this study, HfFG was isolated from the important economical sea cucumber *H. floridana*. Its physicochemical properties, including weight-average molecular weight (Mw) and chemical compositions, were analyzed. To further explore structural characteristics of HfFG, it was selectively cleaved by the deacetylation–deaminative cleavage method. The HfFG fragments were then isolated and purified from the depolymerized products, and the basic chemical structure characteristics of native FG were elucidated by analyzing 1D/2D NMR spectra of the homogeneous oligosaccharide fragments. Furthermore, their effects on the activity of human heparanase were evaluated by the homogeneous time-resolved fluorescence (HTRF) method.

## 2. Results and Discussion

### 2.1. Extraction, Isolation and Purification of HfFG

The yield of purified HfFG was about 8.0 g (~0.9% by dry weight). In this study, the crude polysaccharide was obtained by enzymatic digestion and alkaline treatment, which is the commonly used extraction method of polysaccharide components from sea cucumber [30,31]. The crude polysaccharide was fractionated by the anion-exchange chromatography on a strong anion exchange FPA98 column, and the HfFG component was eluted with 2 M NaCl (HfFG-2M). The HPGPC of HfFG-2M is shown in Appendix A. There were several small peaks, except for the main peak, indicating that HfFG-2M may contain other components with different molecular weight. Further purification was achieved by GPC on a Sepharose CL-6B column. Finally, the HPGPC of HfFG showed a symmetrical peak with elution time at about 14.7 min on a Shodex OH-pak SB-804 HQ column, thus confirming its homogeneity (Figure 1a). Its purity was greater than 99% with a peak area normalization method. In the ^1^H NMR spectrum of HfFG (Figure 2a), the signals at about 1.20–1.30 ppm and 2.0–2.10 ppm could be readily assigned to the methyl protons of fucose and *N*-acetyl-D-galactosamine residues, respectively, with an integral ratio of about 3:2, which is a little higher than that of FGs from many sea cucumber species [32]. The UV spectrum does not show UV absorption near 260 nm or 280 nm, indicating the absence of peptides, proteins or nucleic acids.

### 2.2. Chemical Compositions and Physicochemical Properties of HfFG

The weight-average molecular weight (Mw) of the native HfFG was calculated from a size-exclusion chromatogram obtained using a Shodex OH-pak SB-804 HQ column previously calibrated with standard FG fractions. According to the calibration curve, the Mw of HfFG was estimated to be 47.28 kDa.

Monosaccharide composition analysis results (Figure 1b) showed that HfFG was composed of three types of monosaccharide, including GlcA, GalNAc and Fuc, which is consistent with previous reports of FG from other sea cucumber species [30].

The content of sulfate ester groups is essential for its various biological activities [33]. Thus, the molar ratio of sulfate and carboxyl of HfFG was measured by conductometric titration. The conductivity titration curve is presented in Figure 1c, and it shows two inflexion points, corresponding to the equivalence point of the sulfates and carboxyl groups (COO−), respectively, which is similar to heparin and the FGs reported previously [30,34]. According to the formula, the molar ratio of -OSO_3_^−^/-COO^−^ in HfFG was 3.756. In addition, the specific rotation of HfFG was determined to be −50°.

The IR spectrum of HfFG was shown in Appendix A, and the signal assignments were as follows. The wide and strong absorption band at about 3490 cm^−1^ belonged to the stretching vibration of the OH group. The absorption peak at 2990 cm^−1^ was also observed as the C-H stretching vibration of -CH_3_ on Fuc. 1639 cm^−1^ was the asymmetric stretching vibration of C=O in GlcA and D-GalNAc. The 1415 cm^−1^ absorption peak was the COO^−^ asymmetric stretching vibration in D-GlcA. 1036 cm^−1^ was the stretching vibration of C=O on the sugar ring. In addition, the absorption peaks at 1243 cm^−1^ and 855 cm^−1^ were the stretching vibration of the S=O of the sulfate group and the bending vibration of C-O-S in the sulfate group, respectively.

### 2.3. Preparation and Structure Characteristics of the Low-Molecular-Weight HfFG (dHfFG)

The deacetylated-deaminative depolymerization method could selectively cleave the glycosidic bonds at the GalNAc position to form 2,5-anhydro-D-talose (anTal) residues at the reducing ends of the resulting fragments. It has been used to prepare the depolymerized products of FG from *S. variegatus* and *A. japonicus*, etc. [22,32] These reports on the structure analysis of the depolymerization products revealed that the procedure only breaks the backbone glycosidic bonds and has little effect on other structures of native FG. In the present study, dHfFG was prepared from the native HfFG by deacetylated-deaminative depolymerization with the yields of 60.0%.

The ^1^H NMR spectrum of the partially deacetylated FG is shown in Appendix A, and the degree of deacetylation (DD) was calculated as 78%. The Mw of dHfFG was calculated as 3.90 kDa using a Shodex OH-pak SB-804 HQ column, which is consistent with the theoretical Mw (~3.68 kDa) calculated using DD.

### 2.4. Purification of Homogeneous Oligosaccharides Fragments

HPLC profiles of dHfFG using a Superdex Peptide 10/300 GL column showed that dHfFG was composed of a series of fragments with different sizes (Figure 1d), indicating that they could be purified by repeated gel permeation chromatography (GPC) on Bio-gel P10, P6 and P4 columns. Finally, three homogeneous fragments Fr-1, Fr-2 and Fr-3 were obtained from dHfFG with the yields of 10.9%, 10.2% and 7.1%, respectively. Their HPGPC profiles showed symmetrical peaks (Figure 1d), thus confirming their uniformity.

### 2.5. NMR Analysis of dHfFG, Fr-1, Fr-2 and Fr-3

The NMR spectroscopy has shown great advantages in the clarification of the exact structure of FG compounds. However, the fine structures of native FGs isolated from different species of Holothuria seem to be difficult to interpret, for they usually have spectra with overcrowded signals. The structural analysis of depolymerized fragments was a considerable strategy to give simplified and clear NMR spectra. The structures of HfFG, dHfFG, Fr-1, Fr-2 and Fr-3 were analyzed by NMR spectra.

In the ^1^H NMR spectrum of HfFG (Figure 2a), the signals observed at 1.29 and 1.99 ppm could be assigned to methyl protons of sulfated fucose (FucS) and GalNAc (-(CO)CH_3_) residues, respectively. The broad and overlapping signals at the region of 3.5–5.0 ppm could be assigned to the ring protons and anomeric protons of GalNAc and GlcA. The signals at 5.29, 5.34 and 5.63 ppm could be assigned to the anomeric protons of FucS residues, indicating that HfFG may contain three sulfated types of FucS residues. Unfortunately, due to these signals being overlapped, the ratio of different sulfated types of FucS was hard to determined.

Deaminative cleavage is specific to hexosamine units, and the depolymerization mechanism involves the reaction of nitrous acid with the amine group of the hexosamine unit to form anhydrosugars at the reducing end. In the ^1^H NMR spectrum of dHfFG (Figure 2b), the signal at 4.97 ppm could be readily assigned to the H-1 and H-4 of the 2,5-anhydro-d-talose residues (anTal, T), suggesting the presence of the new anhydrosugars [21]. However, the signals of anomeric and methyl protons of FucS were also overlapped, which hindered the structural analysis of HfFG and dHfFG. Therefore, the three homogeneous fragments Fr-1, Fr-2 and Fr-3 were purified from dHfFG, and their exact structures were studied by 1D and 2D NMR (^1^H-^1^H COSY, TOCSY, ROESY and ^1^H-^13^C HSQC, HMBC) spectra (Figure 3 and Figure 4, Appendix A). The chemical shifts of Fr-1 and Fr-2 shown in Appendix A and Table 1 were based on the interpretations of their 2D NMR spectra, respectively.

The ^1^H NMR spectra of Fr-1, Fr-2 and Fr-3 were similar, except for the differences in the integral area of the corresponding signals (Figure 3a–c). The 1D and 2D NMR spectra of Fr-2 were shown as representatives in Figure 4. The signals at 4.98 ppm and 4.95 ppm could be assigned to H-1 and H-4 of the T residue, according to the literature [22,32], indicating that the new anTal residue at the reducing end was formed. Six distinct signals were observed at the region of 5.2–5.7 ppm, which should be the anomeric protons of fucosyl residues varying in pattern of sulfation, as previously described [22]. According to the ^1^H-^1^H COSY spectrum (Figure 4a), the other H signals of the resonance systems could be further assessed (Table 1). The signals at 5.602, 5.261 and 5.315 ppm were assigned to H-1 of the Fuc_2S4S_ (I’), Fuc_3S4S_ (II’) and Fuc_4S_ (III’) residues linked to GlcA in the middle of the backbone, respectively. Likely, the signals at 5.493, 5.273 and 5.187 ppm could be assigned to H-1 of the Fuc_2S4S_ (I), Fuc_3S4S_ (II) and Fuc_4S_ (III) residues linked to GlcA at the non-reducing termini, respectively. The ratios of different sulfate patterns of FucS could be determined using integral intensities of the respective H-1 signals. The methyl of I’/II’/III’ was observed at 1.24–1.33 ppm, and those methyl protons (I/II/III) were found at 1.1–1.2 ppm (Figure 3a). The integral of these methyl signals could be accurately illustrated by the number of the trisaccharide-repeating units. Thus, Fr-1, Fr-2 and Fr-3 were mainly identified as hexasaccharide, nonasaccharide and dodecasachharide, respectively.

Correspondingly, the carbon data of Fr-2 were analyzed from the ^1^H-^13^C HSQC spectra (Figure 4c). In conclusion, the signal (5.60/99.29 ppm) could be readily assigned to the anomeric H/C of Fuc_2S4S_ residues (I’). According to the superimposed ^1^H-^1^H COSY/TOCSY/ROESY spectra (Figure 4b) of Fr-2, I’ was linked to the C-3 position of the middle GlcA residue (U’). Similarly, the signal (5.49/99.17 ppm) was assigned to H-1/C-1 of the Fuc_2S4S_ residue linked to GlcA (U) at the non-reducing end. The other four sets of signals (5.26/101.85, 5.273/101.55, 5.315/102.04, 5.187/101.19 ppm) were from H-1/C-1 of Fuc_3S4S_ linked to U’ (II’) or U (II) and Fuc_4S_ linked to U’ (III’) or U (III), respectively. These sulfated fucose branches were linked to GlcA residues through α1,3-glycosidic bonds from the analysis of ^1^H-^1^H ROESY spectra. Based on the full assignments of Fr-2 from ^1^H-^13^C HSQC, the GalNAc residues were both sulfated at O-4 and O-6 according to downfield shift carbon (A4 79.05 ppm, A6 69.88 ppm). The cross peaks (U4′, A1) and (A3, U1) in ^1^H-^13^C HMBC spectra (Figure 4d) and the anomeric proton-proton coupling constant (*J_H-H_*) values of GlcA (*J*_1,2_ = 7.2 Hz) and GalNAc_4S6S_ (*J*_1,2_ = 8.8 Hz) indicated that GlcA and GalNAc_4S6S_ residues were linked with alternating β1,3 and β1,4 glycosidic linkages. The data were consistent with the backbones of FGs obtained previously from many other sea cucumber species [22,31].

Based on these analyses, the main chain structure of Fr-1, Fr-2 and Fr-3 was established as d-GlcA-β1,3-{d-GalNAc_4S6S_-β1,4-d-GlcA-β1,3-}*_n_*-d-anTal_4S6S_ (*n* =1, 2, 3), and the FucS were all connected to GlcA by α1,3-glycosidic bonds (Figure 4e). There are three types of sulfation, namely type I (Fuc_2S4S_ ~ 45%), type II (Fuc_3S4S_ ~ 35%) and Type III (Fuc_4S_ ~ 20%). The three types of FucS have no obvious arrangement order.

Therefore, based on the “bottom-up” strategy, it can be reasonably deduced that the structure of the repeating trisaccharide unit of native HfFG was -{4-[l-FucS-α1,3]-d-GlcA-β1,3-d-GalNAc_4S6S_-β1}-, which possessed a CS-E-like backbone and a FucS side chain that was connected to GlcA by the α1,3 glycosidic bonds.

### 2.6. Heparanase Inhibitory Activity

Heparanase activity correlates with the metastatic potential of tumor cells [26]. As a sulfated polysaccharide, FG may be the endogenous substrate structural analogue of heparanase. Therefore, we investigated the effect of FG (HfFG) and its depolymerized fragments on heparanase activity by the HTRF method. Heparin and low molecular weight heparin (LMWH) were used as controls. The results are shown in Figure 5 and Table 2.

The native FG (HfFG) with Mw of 47.28 kDa showed strong inhibitory activity with IC_50_ values of approximately 0.24 nM, which was stronger than that of heparin (1.79 nM). The inhibitory activities of the oligosaccharide components (Fr-1-Fr-3) derived from FG from the sea cucumber *H. floridana* on heparanase were compared, and the IC_50_ range was about 642.3 to 2440 nM. These results indicated that the anti-heparanase activities of FG oligosaccharides decreased with the reduction of its degree of polymerization. Particularly, the activity of Fr-3 was only one-tenth of that of LMWH, while their Mw was 3652 Da and ~4500 Da, respectively. It is worth noting that Fr-1, 2, 3 and LMWH are structurally quite different, since LMWH consists of other monosaccharides, which have other types of glycosidic bonds and sulfate groups. Meanwhile, when compared with the FG oligosaccharides from *H. fuscopunctata* [29], the nonasaccharide (Fr-2, 999.4 ± 138 nM) in our study showed an inhibitory activity similar to that of hendecasaccharide (hs11, 604 ± 38.6 nM), suggesting that the dp of FG oligosaccharides could significantly affect their activities. In addition, the oligosaccharides from *H. fuscopunctata* were highly regular ones, with uniform Fuc_3S4S_ as side chains prepared through the β-eliminative depolymerization method, while the oligosaccharides in this research have complex sulfated fucose (including Fuc_2S4S_, Fuc_3S4S_, Fuc_4S_) and were derived from the deaminative cleavage method. It is worth noting that the difference characteristic of sulfated types of FucS among these oligosaccharides should be important factors on their heparanase inhibitory activities, and it remains to be clarified. Since different depolymerization methods would result in different terminal residues, for instance, and since the β-eliminative depolymerization products have unsaturated uronic acid at the non-reducing end [35] and the deacetylation–deaminative depolymerization products contain a special 2,5-anhydro-d-talose (anTal) residues at the reducing end [32], this structural difference may affect the activity.

Therefore, in addition to the dp, structural characteristics such as the sulfation patterns, the terminal structure of oligosaccharides and the presence of fucosyl branches may also be important factors affecting the heparanase inhibiting activity. The studies on the structure-activity relationship of FG oligosaccharides as heparanase inhibitors still require further systematical analysis.

## 3. Materials and Methods

### 3.1. Materials

The dried body walls of *H. floridana* were purchased from a market of Zhanjiang City, Guangdong province, China. Through the BLAST comparison of the National Bioinformatics Center NCBI gene bank sequence, it was found that the partial sequence of this sea cucumber mitochondrial 16S rRNA gene is 99% similar to *H. floridana* sequence. Therefore, the sea cucumber species was identified as *H. floridana*.

Papain (800 U/mg) was from Shanghai Yuanye Bio-Technology Co., Ltd. (Shanghai, China). Deuterium oxide (D_2_O, 99.9% atom D) was obtained from Sigma-Aldrich (Shanghai, China). l-Rhamnose (Rha), d-galactose (Gal), d-glucose (Glc), d-glucuronic acid (GlcA), and 1-phenyl-3-methyl-5-pyrazolone (PMP) were from Sigma Chemical Co. (St. Louis, MO, USA). D-mannose (Man), l-arabinose (Ara), d-ribose (Rib), d-xylose (Xyl), *N*-acetyl-D-galactosamine (GalNAc) and l-fucose (Fuc) were from Aladdin Chemical Reagent Co., Ltd. (Shanghai, China). Recombinant human heparanase (RhHPSE) was from R & D (Minneapolis, MN, USA). Biotin-heparan sulfate-Eu cryptate, streptavidin-d2 (SA-d2) and HTRF 96-well low volume plates were from Cisbio (Codolet, France). 3-[(3-Cholamidopropyl)-dimethylammonio]-1-propane-sulfonate hydrate (CHAPS), HEPES and potassium fluoride dehydrate (KF) were from Sigma (USA). Amberlite FPA98 Cl ion-exchange resin was purchased from Rohm and Haas Co. Sepharose CL-6B was purchased from GE Healthcare Life Sciences (Uppsala, Sweden). Bio-gel P10, P6 and P4 were purchased from Bio-Rad Laboratories (USA). All other chemicals are analytical grade and available on the market.

### 3.2. Extraction, Isolation and Purification of Polysaccharides

The body walls of *H. floridana* were dried in an oven at 60 °C and then crushed into a uniform powder. The powder (900 g) was treated with 0.1% papain aqueous solution at 50 °C for 6 h, and then treated with 0.5 M sodium hydroxide at 60 °C for 2 h. The protein in the extract was precipitated by adjusting the pH 2.8 with 6 M HCl at 4 °C for 6 h, and removed by centrifugation at 4700 rpm for 15 min. Ethanol was added in the supernatant to a final concentration of 60% (*v*/*v*), and the crude polysaccharide was obtained as precipitate after centrifugation. The crude polysaccharide was further purified by anion exchange chromatography with Amberlite FPA98 ion exchange resin (8 cm × 50 cm), and sequentially eluted with H_2_O, 1.0 M, 2.0 M and 3.0 M NaCl solution. The 2.0 M NaCl fraction was collected, dialyzed (3 kDa cut-off membrane) and then purified by gel permeation chromatography with Sepharose CL-6B (1.5 cm × 150 cm). Finally, the Shodex OH-pak SB-804 HQ column (8 mm × 300 mm) was used for HPLC analysis of the fractions, and the HfFG fractions were combined, dialyzed and lyophilized.

### 3.3. Preparation of Low-Molecular-Weight HfFG by the Deacetylation–Deaminative Cleavage Method

To elucidate the structure of HfFG, its low-molecular-weight products were prepared using partial deacetylation-deamination depolymerization as previously described [21,22]. Briefly, HfFG (2.0 g) was deacetylated in hydrazine monohydrate (500 mL) containing 1% hydrazine sulfate (50 mg) at 90 °C for 27 h. The partially deacetylated HfFG was obtained as white precipitate in ethanol after centrifugation, and then dissolved in distilled water. The resulting solution was dialyzed with a dialysis bag (molecular weight cut-off MWCO 3.5 kDa) and subsequently lyophilized. The degree of deacetylation (DD) was calculated using the integral area ratios of the signals at 1.3 and 2.0 ppm in the ^1^H NMR spectrum (Appendix A), which were from the methyl group of the fucose residue and the GalNAc residue, respectively. Then, the partially deacetylated HfFG (1.34 g) was cleaved with 5.5 M HNO_2_ (pH 4.0) for 10 min in an ice bath, dialyzed with a MWCO of 1000 Da and lyophilized. The yield of dHfFG was 1.2 g.

### 3.4. Purification of Homogeneous Oligosaccharide Fragments from dHfFG

The dHfFG was further fractionated by repeating the gel permeation chromatography (GPC) on the Bio-gel P10 (medium, 2 × 180 cm, Bio-Rad Laboratories, Irvine, CA, USA), Bio-gel P6 (medium, 2 × 180 cm, Bio-Rad) and Bio-gel P4 (fine, 2 × 150 cm, Bio-Rad) columns. Briefly, dHfFG (~0.8 g) was dissolved in 5 mL of deionized water, subjected to the Bio-Gel P10 column equilibrated well with 0.2 M NaCl and then eluted with the same solution. The flow rate was approximately 12 mL/h, and 100 fractions (2 mL/tube) were collected. The absorbance of each fraction was measured by the sulfuric acid-phenol method at 482 nm. The fractions containing the uniform distribution (detected by HPGPC) were combined and then desalted by a Sephadex G-10 (1.5 × 100 cm) column. Finally, three homogeneous fragments from dHfFG were freeze-dried to give Fr-1, Fr-2 and Fr-3.

### 3.5. General Procedures for Physicochemical Properties Determination

The purity and Mw of HfFG and dHfFG were examined by HPGPC using an Agilent 1260 series apparatus (Agilent Technologies, Santa Clara, CA, USA) equipped with a differential refraction detector (RID) and a Shodex OH-pak SB-804 HQ column (8 mm × 300 mm). The elution solvent was 0.1 M NaCl. The flow rate was 0.5 mL/min, and the temperature of the column was 40 °C. For Mw calculation, the Shodex OH-pak SB-804 HQ column (8 mm × 300 mm) was calibrated by a series of fucosylated glycosaminoglycan fractions from *H. fuscopunctata* with Mw of 39.9, 27.8, 14.9, 8.24, 5.30 and 3.12 kDa, determined by a size-exclusion chromatography with multi-angle laser light scattering [36].

The purity of Fr-1, Fr-2 and Fr-3 was examined by HPGPC using an Agilent 1260 series apparatus (Agilent Technologies, Santa Clara, CA, USA) equipped with RID and a Superdex Peptide 10/300 GL column (10 mm × 300 mm). The elution solvent was 0.2 M NaCl. The flow rate was 0.4 mL/min, and the temperature of the column was 35 °C.

The monosaccharide composition of HfFG was analyzed by an Agilent 1260 HPLC system equipped with an Eclipse Plus C18 column (4.6 × 250 mm, 5 µm, Agilent) and a DAD detector. According to the PMP derivatization procedures [14], about 2 mg of HfFG was dissolved in 1 mL of 2 M trifluoroacetic acid (TFA), and then the vessel was sealed and incubated at 110 °C for 4 h in a heating block. Afterwards, the reaction mixture was evaporated to remove the residual TFA with methanol, and the sample was dissolved in 500 µL of H_2_O. Then, 100 μL of the sample solution, 200 μL of 0.5 M PMP in methanol and 100 μL of 0.6 M sodium hydroxide were mixed and incubated at 70 °C for 30 min. After adjusting the pH to 7.0, 500 μL of chloroform was added to extract PMP three times. The top aqueous layer was collected for HPLC analysis.

The sulfate/carboxyl ratio or sulfate group content of HfFG and dHfFG was determined by a classic conductimetric method [34]. About 5 mg of HfFG was dissolved in 1 mL of H_2_O and loaded on a strong cation exchange column (Dowex 50w × 8 50–100 H, 1 × 15 cm) in H^+^ form. It was eluted with 30 mL of distilled water and titrated by 0.02 M NaOH.

The specific rotation of HfFG, dHfFG, Fr-1, Fr-2 and Fr-3 was detected by an Autopol IV automatic polarimeter (Rudolph Research Analytical, Hackettstown, NJ, USA) at a concentration of 1.5 mg/mL at 23 °C. The IR spectra of HfFG, dHfFG, Fr-1, Fr-2 and Fr-3 were measured through a KBr pellet by a Tracer-100 FT infrared spectrometer (Shimadzu, Kyoto, Japan) at the range of 4000–400 cm^−1^.

### 3.6. NMR Analysis

The ^1^H NMR spectra of HfFG, partially deacetylated HfFG and dHfFG were recorded on an AVANCE III TM 600 MHz NMR spectrometer. About 10 mg of each lyophilized sample was dissolved in 0.5 mL of deuterium oxide (D_2_O, 99.9% D) and loaded into a 5-mm nuclear magnetic tube to perform the ^1^H NMR spectrum measurement.

For the 1D/2D NMR spectra measurement of Fr-1, Fr-2, and Fr-3, about 10–15 mg of each sample was prepared by presaturation with D_2_O, then dissolved in 0.5 mL of D_2_O. The ^1^H-^1^H COSY, TOCSY, ROESY, ^1^H-^13^C HMBC and HSQC spectra were recorded at 298 K using standard Bruker pulse sequences on a Bruker Avance 800 MHz spectrometer.

### 3.7. Heparanase Inhibitory Activities Assay

The heparanase inhibitory activities of HfFG, Fr-1, Fr-2 and Fr-3 were determined by a HTRF method [37]. Briefly, 4 μL of sample solution and 3 μL of a heparanase dilution buffer (consisting of 20 mM Tris−HCl (pH 7.4), 0.15 M NaCl and 0.1% CHAPS), were added into 96-well microplates. After pre-incubation at 37 °C for 10 min, the enzyme reaction was initiated by adding 3 μL of Bio-HS-Eu (4.2 ng in 0.2 M NaOAc, pH 5.5) and incubated at 37 °C for 30 min. Then, to detect the remaining substrate, 10 μL of a 1 μg/mL SA-d2 dilution buffer (0.1 M HEPES, 0.8 M KF, 0.1% BSA, pH 7.5) was added. After incubation at room temperature for 15 min, the HTRF signal was detected using a fluorescence reader (Flex Station 3, Molecular Devices), excitation at 340 nm and emissions at 620 nm and 665 nm. Each concentration of the test sample and the reference substance (Heparin or LMWH) was detected four times. The data were fitted by the Origin 2017 software, and the IC_50_ value was expressed as mean ± standard error (SE). The inhibition rate was calculated according to the formula below: Inhibition (%) = (ΔF_sample_ − ΔF_blank_)/(ΔF_max_ − ΔF_blank_)%, and the definition of Δ F was used as reference [34].

## 4. Conclusions

In this study, the structure characterization of HfFG extracted and purified from *H. floridana* was elucidated through a bottom-up strategy. Three homogeneous oligosaccharide fragments derived from the partially deacetylated-deaminative depolymerization method were characterized by 1D/2D NMR spectra. The main structure of native HfFG was shown to contain a typical repeating trisaccharide unit of →4)-[l-FucS-α(1→3)]-d-GlcA-β(1→3)-d-GalNAc_4S6S_-β(1. The GalNAc residues were sulfated both at the O-4 and O-6 positions. HfFG possessed three types of FucS branches (Fuc_2S4S_, Fuc_3S4S_ and Fuc_4S_ with a molar ratio of 45:35:20) linked to O-3 of GlcA through the α1,3 glycosidic linkages.

A heparanase inhibitory activity assay showed that the native FG possessed potent inhibitory activity. The anti-heparanase activities of the three homogeneous oligosaccharides (Fr-1, Fr-2, Fr-3) increased gradually with the increase of molecular weight. The generation of specific heparanase-inhibiting compounds may be a promising approach to develop sulfated polysaccharide-based anticancer lead compounds.

## Figures and Tables

**Figure 1 marinedrugs-19-00162-f001:**
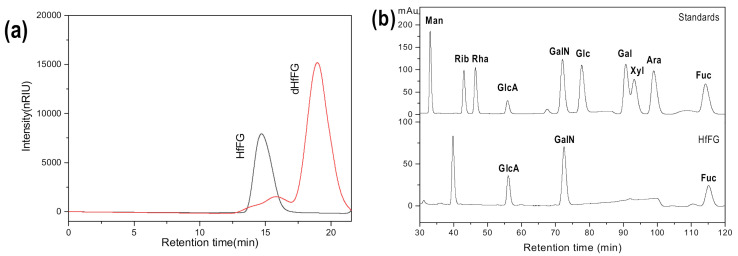
Physicochemical properties of fucosylated glycosaminoglycans isolated from *Holothuria floridana* (HfFG), low-molecular-weight HfFG (dHfFG) and the purified fragments. HPLC profiles of HfFG and dHfFG (**a**) and the oligosaccharide fragments with various degrees of polymerization (dp) (**d**); chromatograms of 1-phenyl-3-methyl-5-pyrazolone (PMP) derivatives of mixed monosaccharide standards and HfFG (**b**); and conductimetric titration curve of HfFG (**c**).

**Figure 2 marinedrugs-19-00162-f002:**
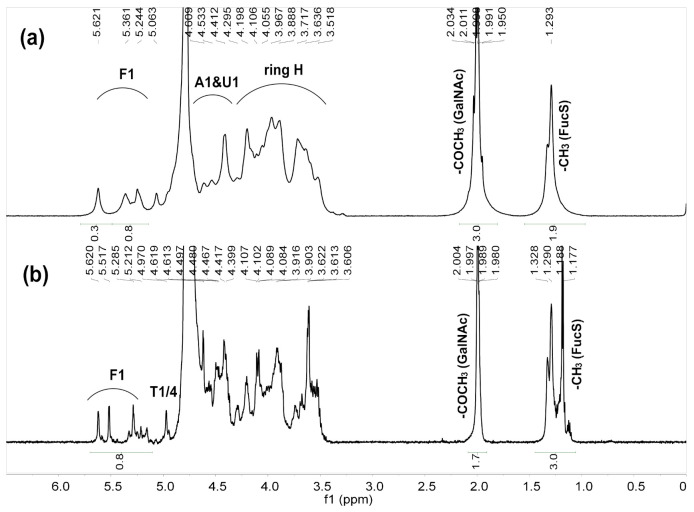
^1^H NMR spectra of HfFG (**a**) and dHfFG (**b**).

**Figure 3 marinedrugs-19-00162-f003:**
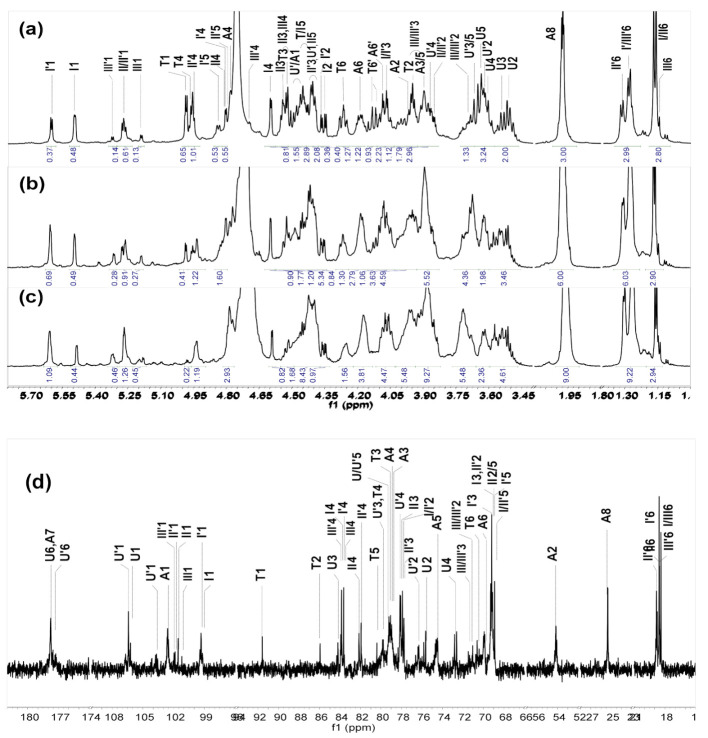
^1^H NMR spectra of Fr-1 (**a**), Fr-2 (**b**), Fr-3 (**c**) and ^13^C NMR spectrum of Fr-2 (**d**).

**Figure 4 marinedrugs-19-00162-f004:**
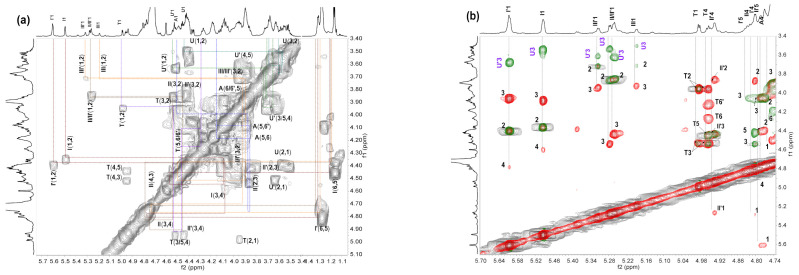
^1^H-^1^H COSY (**a**), overlapped spectra (**b**) of ^1^H-^1^H COSY (**gray**), ROESY (**green**) and TOCSY (**red**), ^1^H-^13^C HSQC (**c**) and partial ^1^H-^13^C HMBC (**d**) spectra of Fr-2 and the chemical structure of native HfFG and its oligosaccharides (**e**). Fuc_2S4S_, Fuc_3S4S_ and Fuc_4S_ were signed as I, II, III, respectively, and U, A, T represent the GlcA, GalNAc and anTal residues, respectively.

**Figure 5 marinedrugs-19-00162-f005:**
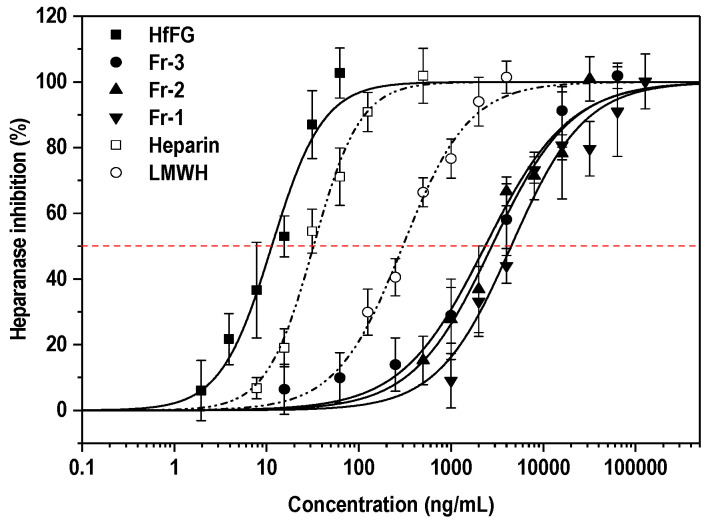
Inhibition effects of HfFG and its oligosaccharides on heparanase (mean ± SD, *n* = 4).

**Table 1 marinedrugs-19-00162-t001:** ^1^H/^13^C NMR chemical shift assignments of Fr-2 (800 MHz, D_2_O).

Residue.		Chemical Shift (ppm)								
		H/C	1	2	3	4	5	6	7	8
**T**	→3)-β-d-anTal_4S6S_-diol	H	4.983	3.949	4.523	4.952	4.433	4.263/4.119		
		C	91.58	85.93	79.13	79.79	80.38	70.6		
**U’**	→4)-β-d-GlcA-(1→	H	4.484	3.623	3.673	3.893	3.673			
		C	103.64	76.38	79.79	78.13	79.32	177.34		
**A**	→3)-β-d-GalNAc_4S6S_-(1→	H	4.485	3.971	3.938	4.729	3.893	4.101/4.184		
		C	102.58	54.05	78.9	79.05	74.53	69.88	177.76	25.3
**F’ (I’)**	α-l-Fuc_2S4S_-(1→	H	5.602	4.403	4.064	4.766	4.817	1.276		
		C	99.3	77.88	67.27	83.62	69.04	18.53		
**F’ (II’)**	α-l-Fuc_3S4S_-(1→	H	5.261	3.857	4.422	4.942	4.783	1.303		
		C	101.85	69.18	77.75	81.92	68.89	18.8		
**F’(III’)**	α-l-Fuc_4S_-(1→	H	5.315	3.708	3.949	4.69	4.745	1.267		
		C	102.04	71.33	71.46	83.94	68.98	18.47		
**U**	d-GlcA-(1→	H	4.416	3.512	3.551	3.618	3.671			
		C	106.4	76.63	84.19	72.6	79.32	177.76		
**F (I)**	α-l-Fuc_2S4S_-(1→	H	5.493	4.357	4.079	4.596	4.452	1.167		
		C	99.17	77.88	69.18	83.83	68.89	18.35		
**F (II)**	α-l-Fuc_3S4S_-(1→	H	5.273	3.856	4.437	4.933	4.769	1.167		
		C	101.55	69.1	78.06	82.13	69.1	18.5		
**F (III)**	α-l-Fuc_4S_-(1→	H	5.187	3.711	3.93	4.511	4.408	1.144		
		C	101.19	71.33	71.46	83.55	68.98	18.35		

**Table 2 marinedrugs-19-00162-t002:** Heparanase inhibitory activities of various samples.

Sample	Mw (Da)	Specific Rotation [α]	IC_50_ (ng/mL)	IC_50_ (nM)
HfFG	47,282	−50°	11.4 ± 1.39	0.24 ± 2.94
Fr-3	3652 ^a^	−45°	2346 ± 409	642.3 ± 112
Fr-2	2767	−32°	2766 ± 381	999.4 ± 138
Fr-1	1841	−44°	4496 ± 590	2440 ± 320
Heparin	~18,000	--	32.3 ± 2.21	1.79 ± 0.12
LMWH	~4500	--	300 ± 29.4	66.67 ± 6.53

^a^ The Mw of Fr-1, Fr-2 and Fr-3 was calculated by ChemBioDraw Ultra 14.0 software based on the degree of polymerization and the composition of FucS from the integral analysis of their ^1^H NMR spectra, respectively.

## Data Availability

The data presented in this study are available on request from the corresponding author without restriction.

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
