# Peer review of "Structural Characterization and Heparanase Inhibitory Activity of Fucosylated Glycosaminoglycan from Holothuria floridana"

_marinedrugs, 2021, doi:10.3390/md19030162_

Round 1

Reviewer 1 Report

The current manuscript entitled “Structural characterization and heparanase inhibitory activity of fucosylated glycosaminoglycan from Holothuria floridana” (Shi X et al., Manuscript ID: marinedrugs-1138135) describes structural characterization of GAG from sea cucumber Holothuria floridana using 1D and 2D NMR spectroscopy technique. Additionally, the authors newly discovered heparanase inhibitory activity in the polysaccharide. A series of the authors’ researches continuously and profoundly examine this type of GAGs.

Major Comments
1) The current study uses Holothuria floridana. In a precedent paper, a part of the authors used a similar material Holothuria coluber (Carbohydr Polym 233:115844, 2020). It is likely that the two fucosylated GAGs share the same core structure -{(L-FucS-α1,3-)D-GlcA-27 β1,3-D-GalNAc4S6S-β1,4-}-. Are there any structural diversity in fucosylated GAGs prepared from different species, Holothuria floridana and Holothuria coluber, and other species? It is not likely that the present manuscript include discussion whether the current fucosylated GAGs are structurally novel or not.
2) A related article has been recently published (Oligosaccharides from fucosylated glycosaminoglycan prevent breast cancer metastasis in mice. Pharm Res Available online 2 March 2021, 105527). To avoid any possible duplicate submission, the authors should cite the paper and explain a novel point of the current study that is not included in the paper of Pharm Res. To reinforce the indicated heparanase inhibitory activity, it is strongly recommended to show evidence that the fucosylated GAGs directly bind to heparanase molecule.

Author Response

Point 1: The current study uses Holothuria floridana. In a precedent paper, a part of the authors used a similar material Holothuria coluber (Carbohydr Polym 233:115844, 2020). It is likely that the two fucosylated GAGs share the same core structure -{(L-FucS-α1,3-)D-GlcA-27 β1,3-D-GalNAc4S6S-β1,4-}-. Are there any structural diversity in fucosylated GAGs prepared from different species, Holothuria floridana and Holothuria coluber, and other species? It is not likely that the present manuscript include discussion whether the current fucosylated GAGs are structurally novel or not. 

Response 1: Thanks for the reviewer’s question,this is a very good question.

In general, as a bioactive polymer, structural variations of FG focus on the amount and position of branches, as well as on degree and pattern of sulfation of backbone and branches. The sulfated fucose (FucS) branches were sulfated at different positions in different proportions, varying from the sea cucumber species (Pomin. Mar. Drugs. 2014, 12, 232-54; Fan et al. Yao Xue Xue Bao. 1980, 263-270. Ustyuzhanina, et al, Pure Appl. Chem. 2019, 91, 1065-1071).

Therefore, although both HcFG and HfFG have a CS-E-like backbone and a FucS side chain, the sulfated degree and proportion of FucS are significantly different (The side chain FucS in HcFG is Fuc2S4S, Fuc3S4S and Fuc4S, with a ratio of 37:50:13; the side chain FucS in HfFG is Fuc2S4S, Fuc3S4S and Fuc4S, with a ratio is 45:35:20) .The research about the FG from Holothuria coluber has been cited as Reference 24 in our revised manuscript.

Point 2: A related article has been recently published (Oligosaccharides from fucosylated glycosaminoglycan prevent breast cancer metastasis in mice. Pharm Res Available online 2 March 2021, 105527). To avoid any possible duplicate submission, the authors should cite the paper and explain a novel point of the current study that is not included in the paper of Pharm Res. To reinforce the indicated heparanase inhibitory activity, it is strongly recommended to show evidence that the fucosylated GAGs directly bind to heparanase molecule.

Response 2: Thanks for the reviewer’s valuable comments, which will be of great help to my work. According to the reviewer’s suggestion, the comparison and discussion about the heparanase inhibitory activities of native HfFG and oligosaccharide fragments has been improved in the revised manuscript (Line 274-293). The related article has been cited as Reference 29 in our revised manuscript. Since Fr-1, Fr-2 and Fr-3 were oligosaccharides composed of a series of hexasaccahrides, nonasaccharides and dodecasachharide with various FucS branches respectively, the direct binding affinity of Fr-1, Fr-2 and Fr-3 to heparanase has not been measured. The oligosaccharide with defined sequence will be further purified and the directly bind affinity to heparanase will be evaluated using surface plasmon resonance (SPR) assay, which has been proved useful in our recently study.

Reviewer 2 Report

The paper of Xiang Shi et al. describes the structural characterization and heparanase inhibitory activity of fucosylated glycosaminoglycan and oligosaccharides from Holothuria floridana. The structure was characterized using appropriate methods, and conclusions are sound. In my opinion, the work lacks discussion with previous work devoted to research fucosylated glycosaminoglycan from Holothuria floridana (Shi et.al, 2019, International Journal of Biological Macromolecules, 132, 738-747). Using a different depolymerization method, the authors confirmed the proposed structure. There are many methods of depolymerization, so the question arises about the novelty of the study (Li et.al, 2021, Carbohydrate Polymers, 251, 117034). The advantages of the work include the heparanase inhibitory activity. I have no fundamental comments on this work, but in my opinion, the novelty of the study is not sufficient for publication.

Reviewer 3 Report

This is an interesting paper focused on the structural characterization of a fucosylated glycosaminoglycan from Holothuria floridana and its Heparanase Inhibitory Activity.

The authors isolated the high MW polymer and depolymerize it according to a deacetylation deamination procedure, which is a very good trick. After this they carry out an extensive and ample structural characterization (essentially but not only) by NMR. After this they test the three oligoes and establish that the high MW but also the fucose and sulfate substitution do have a role in the polymer activity.

I am full satisfied of such a good paper and such a work.

Author Response

Thank you very much for your recommendation of our manuscript.

Round 2

Reviewer 1 Report

The authors properly responded to the reviewer's comments.  

Author Response

Point 1: The authors properly responded to the reviewer's comments.

Response 1: Thank you very much for your recommendation of our manuscript.

Reviewer 2 Report

I have no fundamental comments

Author Response

Point 1: I have no fundamental comments.

Response 1: Thank you very much for your recommendation of our manuscript.

This manuscript is a resubmission of an earlier submission. The following is a list of the peer review reports and author responses from that submission.